# Dark matter from axion strings with adaptive mesh refinement

Malte Buschmann[1✉], Joshua W. Foster[2,3,4✉], Anson Hook[5], Adam Peterson[6], Don E. Willcox[6], Weiqun Zhang[6] & Benjamin R. Safdi [3,4✉]

Axions are hypothetical particles that may explain the observed dark matter density and the non-observation of a neutron electric dipole moment. An increasing number of axion laboratory searches are underway worldwide, but these efforts are made difficult by the fact that the axion mass is largely unconstrained. If the axion is generated after inflation there is a unique mass that gives rise to the observed dark matter abundance; due to nonlinearities and topological defects known as strings, computing this mass accurately has been a challenge for four decades. Recent works, making use of large static lattice simulations, have led to largely disparate predictions for the axion mass, spanning the range from 25 microelectronvolts to over 500 microelectronvolts. In this work we show that adaptive mesh refinement simulations are better suited for axion cosmology than the previously-used static lattice simulations because only the string cores require high spatial resolution. Using dedicated adaptive mesh refinement simulations we obtain an over three order of magnitude leap in dynamic range and provide evidence that axion strings radiate their energy with a scale-invariant spectrum, to within ~5% precision, leading to a mass prediction in the range (40,180) microelectronvolts.

[1] Department of Physics, Princeton University, Princeton, NJ 08544, USA. [2] Leinweber Center for Theoretical Physics, Department of Physics, University of Michigan, Ann Arbor, MI 48109, USA. [3] Berkeley Center for Theoretical Physics, University of California, Berkeley, CA 94720, USA. [4] Theoretical Physics Group, Lawrence Berkeley National Laboratory, Berkeley, CA 94720, USA. [5] Maryland Center for Fundamental Physics, University of Maryland, College Park, MD 20742, USA. [6] Center for Computational Sciences and Engineering Lawrence Berkeley National Laboratory, Berkeley, CA 94720, USA.
✉email: msab@princeton.edu; jwfoster@mit.edu; brsafdi@berkeley.edu

An outstanding mystery of the Standard Model of particle physics is that the neutron electric dipole moment, which would cause the neutron to precess in the presence of an electric field, appears to be over ten billion times smaller than expected[1]. Axions were originally invoked as a dynamical solution to this problem; they would interact with quantum chromodynamics (QCD) inside of the neutron so as to remove the electric dipole moment[2–5]. However, free-streaming ultra-cold axions may also be produced cosmologically in the early Universe, and these axions may explain the observed dark matter (DM)[6–8], which is known to govern the dynamics of galaxies and galaxy clusters.

Multiple efforts are underway at present to search for the existence of axion DM in the laboratory[9,10], but these efforts are hindered by the fact that the mass of the axion particle is currently unknown. The axion is naturally realized as the pseudo-Goldstone boson of a global symmetry called the Peccei-Quinn (PQ) symmetry, which is broken at a high energy scale $f_a$[2–5,11]. If the PQ symmetry is broken after the cosmological epoch of inflation, then there is a unique axion mass $m_a$ that leads to the observed DM abundance. (If the PQ symmetry is broken before or during inflation, then the DM abundance depends on the initial value of the axion field that is inflated[12].) However, computing this mass is difficult principally because after PQ symmetry breaking axion strings develop; at the string cores the full PQ symmetry is restored. As the Universe expands the strings shrink, straighten, and combine by emitting radiation into axions. The contribution to the DM abundance from the string-induced axions has been heavily debated, with some works claiming that string-induced axions play a minor role[13,14], with the DM abundance dominated by axions produced during the QCD phase transition, and others claiming these axions dominate the DM abundance[15–17]. The evolution of the axion string network in the early Universe has been studied numerically and analytically since the 1980's[13–22] with increasingly complex and capable frameworks in recent years[23–28]. The earliest simulations were restricted computationally to lattices of order ~$150^3$ sites[15], while modern-day static-lattice simulations have achieved ~$8,000^3$ sites[25].

In this work, we use adaptive mesh refinement (AMR) simulations to provide an even larger jump in sensitivity by maintaining high resolution around the string cores and lower resolution elsewhere[29]; to achieve the same resolution as our simulations using a static grid would require a $65,536^3$ site lattice. Our substantial dynamical range allows us to determine that radiation from axion strings prior to the QCD phase transition likely dominates the DM density.

## Results

### AMR simulation framework.
The axion $a$ is the phase of the complex PQ scalar field $\Phi = (r + f_a)/\sqrt{2}e^{ia/f_a}$, with $a = a(x)$ and $r = r(x)$ real functions of spacetime $x$. The radial mode $r$ is heavy and is not dynamical at temperatures below its mass $m_r$. The axion field, on the other hand, is massless until the QCD phase transition and thus is dynamical on scales smaller than the cosmological horizon between the PQ and QCD epochs. The axion field acquires a small mass $m_a \sim \Lambda_{QCD}^2/f_a$ at temperatures $T$ of order the QCD confinement scale $\Lambda_{QCD}$ from QCD instantons[30], though in our simulations we focus on temperatures $T \gg \Lambda_{QCD}$ where the mass may be neglected.

Our simulation is based on the block-structured AMR software framework AMReX[31]. The equations of motion (EOM) for $\Phi$ can be derived from the Lagrangian[32]

$$\mathcal{L}_{PQ} = |\partial\Phi|^2 - \lambda\left(|\Phi|^2 - \frac{f_a^2}{2}\right)^2 - \frac{\lambda T^2}{3}|\Phi|^2, \qquad (1)$$

where $\lambda$ is the PQ quartic coupling. (We fix $\lambda = 1$ without loss of generality so that $m_r = \sqrt{2}f_a$.) The EOM is solved using the strong-stability preserving Runge-Kutta (SSPRK3) algorithm with a time step size that satisfies the Courant-Friedrichs-Lewy condition on a lattice defined in fixed comoving coordinates. Evolution takes place in rescaled conformal time $\eta = R/R_1 = (t/t_1)^{1/2}$, where $R$ is the scale factor of the Friedmann-Lemaître-Robertson-Walker metric, $R_1 \equiv R(t_1)$, and $t_1$ is a reference time such that $H_1 \equiv H(t_1) = f_a$ with Hubble parameter $H$. In these units the PQ phase transition takes place around $\eta \approx 1$, and we chose a starting time of $\eta_i = 0.1$ and a final time of $\eta_f = 75.7$. Our simulation volume is a box with periodic boundary conditions and comoving side length $L = 120/(R_1H_1)$. This volume corresponds to a box length of 1200 Hubble lengths at $\eta_i$ and ~1.6 Hubble lengths at $\eta_f$. (Gorghetto et al.[27] found that finite-volume effects are not important for simulations ending with a box length of ~1.5 Hubble lengths).

The string width scale $\Gamma$ is set by $m_r^{-1}$, while the maximum physical length scale that may be resolved with the comoving lattice grows linearly with $\eta$. Thus, finer grids are needed to resolve $\Gamma$ at later times. We start with a uniform grid of $2048^3$ grid sites, with an initial state based on a thermal distribution before the PQ phase transition (see Methods). Extra refined grids are then added over time whenever the comoving string width drops below a certain threshold. We add the first four extra refinement levels when $\Gamma$ is resolved by four grid sites at the respective finest level, with the fifth extra level added when $\Gamma$ is resolved by three grid sites (see Fig. 1 and Supplementary Fig. 1). In comparison, note that Gorghetto et al.[27] resolves $\Gamma$ by one grid site at the end of their simulation. Each extra level introduces eight times as many grid cells per volume as the previous level. Refined levels are localized primarily around strings. This is achieved by identifying grid cells that are pierced by a string core using the algorithm described in Fleury and Moore[33]. The exact grid layout is periodically adjusted to track strings over time. See Fig. 1 for an illustration of the grid layout. Convergence tests of the AMR framework and tests of the dependence on the initial state are described in Methods section.

### String network evolution.
The axion string network is thought to evolve and shrink with time by radiating axions so as to obey the scaling solution, where the number of strings per Hubble patch remains order unity as a function of time[20]. The network evolution is illustrated in the top panels of Fig. 1, with time slices labeled by $\log(m_r/H) = \log(2m_rt)$. The energy density in axion radiation is overlaid on top of the string network and is strongest in the vicinity of areas of large string curvature.

The string length per Hubble volume is quantified through the parameter $\xi$, which is defined by $\xi \equiv \ell t^2/\mathcal{V}$ with $\ell$ the total string length in the simulation volume $\mathcal{V}$. We determine $\ell$ by counting string-pierced plaquettes in our simulation using the algorithm described in Fleury and Moore[33]. We illustrate $\xi$ as a function of $\log m_r/H$ in Fig. 2.

We compute $\xi$ at points in time separated by a Hubble time ($\Delta \log m_r/H = \log 2$), since the network is strongly correlated on time scales smaller than a Hubble time.

We verify that $\xi$ increases linearly with $\log m_r/H$, which was first suggested in refs.[24,27]. Gorghetto et al.[27] constructed a suite of simulations on static grids of up to $4500^3$ sites and out to at most $\log m_r/H \sim 7.9$; they fit a model of the form $\xi = c_{-2}/\log^2 + c_{-1}/\log + c_0 + c_1\log$, with $\log \equiv \log m_r/H$, to their $\xi$ data for $\log \in (4.5, 7.9)$ and found $c_1 = 0.24 \pm 0.02$ for the leading term. Given $m_r \sim 10^{10}$ GeV and the QCD phase transition beginning at temperatures $T \sim 1$ GeV, the string network is expected to evolve until $\log_* \sim 65$, which is far beyond the dynamical range that may be simulated.

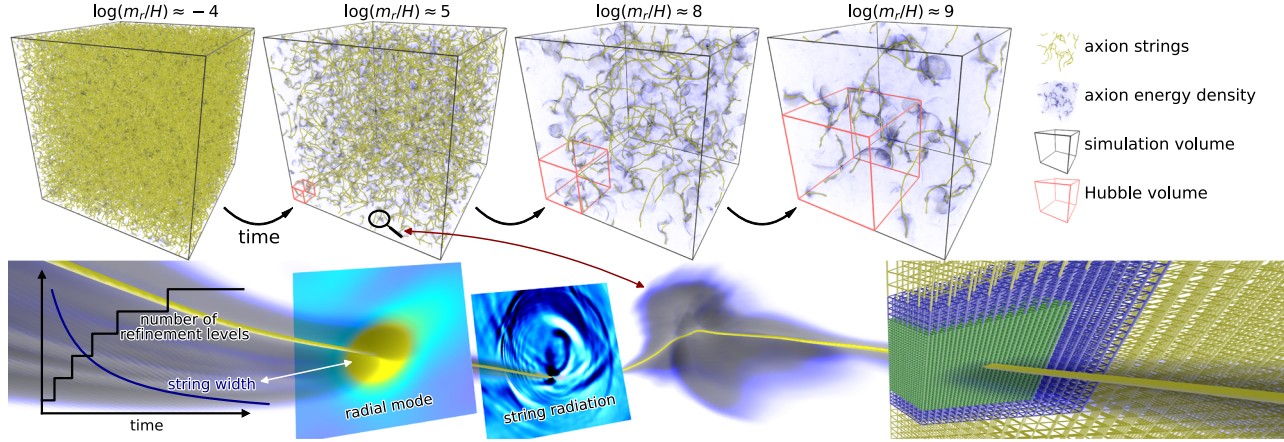

**Fig. 1 Simulation snapshots and illustrations.** (Top row) 3-D rendering of various simulation states from the initial state (left) to the final state (right). Shown is the full simulation volume with the respective relative size of a Hubble volume indicated. The axion energy density is illustrated by the density of a 3-D media and string cores are overlaid in yellow. (Bottom row) Zoom in on a string segment. From left to right: Relationship between the string width and the number of refinement levels as a function of time; 2-D slices of the radial mode and string radiation centered around a string element; string element enshrouded by axion energy density; and an illustration of the layout of the three coarsest grid levels around a string core (not to scale). Animations available here and can be downloaded at Buschmann et al.[54].

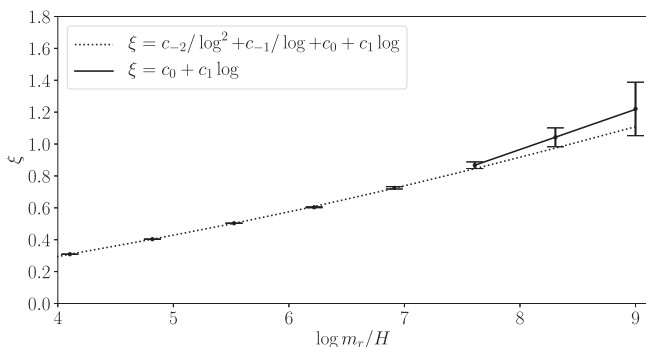

**Fig. 2 Evolution of the string length.** The string length per Hubble volume $\xi$ increases with time in our simulation, indicating a logarithmic violation to the scaling solution[24], which would predict constant $\xi$. At late times in the simulation (large $\log m_r/H$) the growth in $\xi$ appears linear in $\log m_r/H$ with coefficient $c_1 \approx 0.25$ as measured for the fit over the full $\log m_r/H$ range shown, but including terms all the way down to $c_{-2}/\log^2$. The fit illustrated by the solid curve only includes terms down to $c_0$ but is limited to late times ($\log \in (7.5, 9)$); this fit leads to $c_1 \approx 0.25$ also. These fits indicate that at the beginning of the QCD phase transition, at $\log_* \approx 65$, we expect $\xi_* \approx 15$. The error bars in $\xi$ correspond to normally distributed 68% confidence intervals derived during the fitting procedure.

In Fig. 2 we illustrate our fit of the same functional form as in Gorghetto et al.[27] to our $\xi$ data over the range $\log \in (4, 9)$; we find $c_1 = 0.254 \pm 0.002$ (see Methods section for details). As a systematic test we fit the functional form $\xi = c_0 + c_1 \log$ to the $\xi$ data over the limited range $\log \in (7.5, 9)$ and determine $c_1 \approx 0.252$. Importantly, the parameter $c_1$, which governs the large log behavior of $\xi$, agrees between the two methods and agrees with the measurement in Gorghetto et al.[27]. Assuming that the QCD phase transition begins at $\log_* \in (60, 70)$ we estimate that at the beginning of the phase transition $\xi = \xi_* \in (13, 17)$. The linear growth of $\xi$ with $\log m_r/H$ does not support the analytic velocity-dependent one-scale model (see refs. [34–36]), which predicts that $\xi$ should approach a constant at large log. On the other hand, the observation that $\xi$ grows linearly with log may be naturally explained by the well-established logarithmic increase of the string tension with time, $\mu(t) \approx \mu_0 \log m_r/H$ with $\mu_0 = \pi f_a^2$ to leading order in large log (see Supplementary Fig. 2). A given

string segment loses energy at a constant rate that does not evolve with time[20], and as a result energy builds up in the strings relative to the situation where $\mu$ does not increase logarithmically with time. This increase in energy is manifest by a logarithmically increasing $\xi$. (See Methods section for details of this argument.)

**Axion radiation spectrum.** As the string network evolves in the scaling regime axions are produced at a rate $\Gamma_a \approx 2H\rho_s$, where $\rho_s = \xi\mu/t^2$ is the energy density in strings. As we show later in this Article, the DM density from string-induced axion radiation is proportional to the number density of axions at $\log m_r/H = \log_*$. To compute the number density we need to know the axion radiation spectrum from strings. We quantify the spectrum through the normalized distribution $F(k/H) = d \log \Gamma_a/d(k/H)$ for physical momentum $k$. (See, e.g., Gorghetto et al.[24] for a review of the analytic aspects of the network evolution.) We compute $F$ numerically from our simulation ouput by $F(k/H) \propto (1/R^3) \frac{d}{dt}(R^3 \partial\rho_a/\partial k)$, with $\partial\rho_a/\partial k$ the time-dependent differential axion energy density spectrum.

The axion radiation is distributed in frequency between the effective infrared (IR) cutoff, which is provided by $H$, and the effective ultraviolet (UV) cutoff set by the string width $\sim m_r$. For momenta $k$ well between these two scales ($H \ll k \ll m_r$) the radiation spectrum is expected to follow a power-law. Below, we describe how we measure the index of this power law.

We calculate $F$ via finite differences in nonuniform $\Delta t$ corresponding to uniform intervals in $\log m_r/H$. In our fiducial analysis, we calculate instantaneous emission spectra using intervals of $\Delta \log m_r/H = 0.25$, which is of order Hubble-time separations. At each $\log m_r/H$ value, we fit a power-law model $F(k/H) \propto 1/(k/H)^q$ to the instantaneous spectra between an IR cut-off $k_{IR} = x_{IR}H$ and a UV cut-off $k_{UV} = m_r/x_{UV}$, with the cut-offs chosen to be sufficiently far from the physical IR and UV cut-offs. (See the methods for details of how this fit is performed.) We chose $x_{IR} = 50$ and $x_{UV} = 16$ in order to be sufficiently far into the power law regime of $k$.

In the top panel of Fig. 3 we illustrate $F$ computed at $\log m_r/H = 8.75$ for our fiducial choice of $x_{IR}$ and $x_{UV}$ as well as two systematic variations on the choice of fitting range, extending to $x_{IR} = 30$ ("Extended IR Data") and $x_{UV} = 12$ ("xtended UV Data"). The best-fit power-law models are also illustrated. In the bottom panel, we show the evolution of the index $q$ as a function

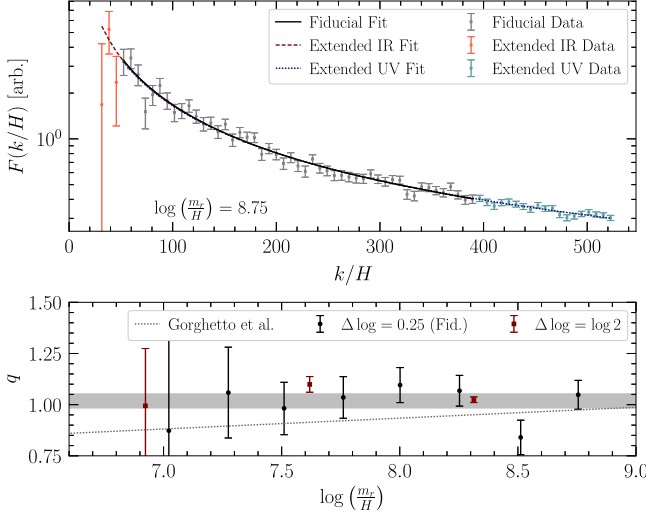

**Fig. 3 The instantaneous emission spectra.** (Above) Example fits to the instantaneous emission spectrum calculated at $\log m_r/H = 8.75$. In our fiducial analysis, the instantaneous emission spectra are calculated using a timestep corresponding to $\Delta \log m_r/H = 0.25$, and a power-law model is fit to the data at $k$ between the IR and UV cutoffs of $k_{\mathrm{IR}} = 50H$ and $k_{\mathrm{UV}} = m_r/16$. The data included in this fit range is shown in gray with the best-fit power law depicted in black; the indicated uncertainties are at 68% confidence and are derived during the fitting procedure. We also illustrate two systematic variations, one in which we extend our IR cutoff down to $k_{\mathrm{IR}} = 30H$ ("Extended IR Data"), and another where we extend our UV cutoff upward to $k_{\mathrm{UV}} = m_r/12$ ("Extended UV data"). For clarity, the data are down-binned by a factor of 2 in $k/H$. (Below) The evolution of the fitted power-law index $q$ as a function of $\log m_r/H$. The best fit indices obtained in our fiducial analysis are shown in black, with red showing the indices computed using $\Delta \log m_r/H = \log 2$. In our fiducial analysis we constrain $q = 1.02 \pm 0.03$, which is shaded. For comparison, the best fit linear growth of $q$ obtained in Gorghetto et al.[27] is shown in dotted gray.

of $\log m_r/H$, for both our fiducial analysis and for a systematic variation where we use $\Delta \log m_r/H = \log 2$ when computing $F$. We compare our results to the best-fit model obtained in Gorghetto et al.[27], who claimed evidence that $q$ evolves logarithmically in time, with $q > 1$ at late times. In particular, Gorghetto et al.[27] fit the evolution model $q(t) = q_1 \log(m_r/H) + q_0$ to their $q$ data and found evidence for non-zero $q_1$, claiming $q_1 = 0.053 \pm 0.005$. Fitting this model to our $q$ data (see Methods section for details) yields $q_1 = -0.04 \pm 0.08$ and $q_0 = 1.36 \pm 0.69$, which is in tension with the results in Gorghetto et al.[27]. (The best-fit model in that work is inconsistent at the level $\sim 1.8\sigma$ with our measured $q$ values). Given that we do not find evidence for logarithmic growth of $q$, we impose $q_1 = 0$ and find $q_0 = 1.02 \pm 0.04$, which is interestingly consistent with the scale invariant spectrum $q_0 = 1$, suggested in refs. [13,28], to within ~5%. An additional argument in favor of $q_0 = 1$ is that the string loops appear logarithmically distributed in size, as shown in Supplementary Fig. 3 and as expected for a network of intersecting strings (see Methods section).

One difference between Gorghetto et al.[27] and this work that may contribute to the difference in $q$ is that Gorghetto et al.[27] used $x_{\mathrm{UV}} = 4$; in Supplementary Fig. 4 we show that using $x_{\mathrm{UV}} = 4$ in our fits also leads to positive $q_1$ at non-trivial significance (see Supplementary Tab. II); however, as illustrated in Supplementary Fig. 8 at large $\log m_r/H$ and $x_{\mathrm{UV}} = 4$ the fits become visibly poor at large $k/H$ because the spectrum $F(k/H)$ begins to drop rapidly for $k \sim m_r$. The fact that Gorghetto et al.[27] is only resolving the string cores by around one grid site at large

$\log m_r/H$ may also play a role. We test the importance of the string-core resolution by performing an alternate simulation where we do not add extra refinement levels after $\log m_r/H \approx 5.3$, such that $\Gamma$ is resolved by one grid site at $\log m_r/H \approx 8.1$ (see Supplementary Fig. 1). As illustrated in Supplementary Fig. 10, in this case the spectrum becomes distinctly biased towards larger $q$ at larger log, where the string-core resolution is low.

Our result that $q_1$ is consistent with zero is robust to changes to $x_{\mathrm{UV}}$ (Supplementary Fig. 4 and Supplementary Tab. II), for $32 \gtrsim x_{\mathrm{UV}} \gtrsim 8$, to $x_{\mathrm{IR}}$ (Supplementary Fig. 5 and Supplementary Tab. I), for the range $30 \leq x_{\mathrm{IR}} \leq 100$ that we consider, to the $\Delta \log$ size used in computing $F$ (Supplementary Fig. 6 and Supplementary Tab. III), for $0.125 \leq \Delta \log \leq \log 2$, and to the method used for regulating the string cores when computing $F$ (Supplementary Fig. 7 and Supplementary Tab. IV).

**Dark matter density.** The axion EOM during the QCD epoch generically violates number density conservation. In particular, the non-linear axion potential is a function of $\cos(a/f_a)$, which implies that non-linear terms in the EOM are important if $|a/f_a| \gtrsim \pi$. Given the instantaneous spectrum $F(k/H)$ we may compute the average field value squared at a given time $t$ by $\langle (a/f_a)^2 \rangle \approx 4\pi \int^t (dt'/t) \xi(t') \langle (H'/k')^2 \rangle \log m_r/H'$, with $\langle (H'/k')^2 \rangle$ being the expected value of $H/k$ at time $t'$ computed from the distribution $F(k/H)$ (see Methods section and note that this is accurate to leading order in $\log m_r/H$). We expect $\langle (H/k)^2 \rangle$ to be proportional to $H^2/k_{\mathrm{IR}}^2$, with $k_{\mathrm{IR}}/H \propto \sqrt{\xi}$ being the effective IR cut-off for $F(k/H)$ that arises from the typical separation of strings $\sim k_{\mathrm{IR}}^{-1}$; note that this implies that as $\xi(t)$ grows with time, the effective IR cut-off moves towards the UV like $\sqrt{\xi}$ because the strings become more closely packed together. Let us define a dimensionless coefficient $\beta$ by the relation $\langle (H/k)^2 \rangle^{-1} = \beta \xi$; a fit of this functional form to the spectral data leads to $\beta = 840 \pm 70$ for $q = 1.06$ (see Supplementary Fig. 9). Note that smaller values of $q$ lead to larger values of $\beta$ and that $q = 1.06$ is the maximum value of $q$ allowed at $1\sigma$ from our analysis. In terms of this coefficient $\langle (a/f_a)^2 \rangle \approx (4\pi/\beta) \log m_r/H \lesssim 1.1$ (for $\log m_r/H \lesssim 70$), which implies that non-linear number changing processes are at most marginally relevant. (Non-linear corrections to the linearized force are at most $\sim 15\%$.) This justifies our use of number density conservation below in estimating the DM abundance.

To compute the axion number density we need to compute the expectation value $\langle H/k \rangle$ over the distribution $F(k/H)$. Following the justification in the previous paragraph we may parameterize this expectation value in terms of the IR cut-off and thus $\xi$, $\langle H/k \rangle^{-1} = \delta \sqrt{\xi}$, for a dimensionless parameter $\delta$. In Fig. 4 we illustrate the $\langle H/k \rangle^{-1}$ data, assuming $q = 1.06$, as a function of $\log m_r/H$ along with the best fit model, which leads to $\delta = 113 \pm 7$; note that smaller values of $q$ lead to larger values of $\delta$. To compute $\langle H/k \rangle^{-1}$ (and also $\langle (H/k)^2 \rangle^{-1}$) we numerically integrate the spectrum up to $k/H = x_{\mathrm{max}}$, with $x_{\mathrm{max}} = 50$, and then analytically integrate the power-law functional form $F(k/H) \propto 1/k^q$ from $x_{\mathrm{max}}$ to $k/H \sim e^{\log_*}$, with $\log_* \sim 60 - 70$. The axion number density at the epoch of the QCD phase transition is then, to leading order in $\log_*$, $n_a^{\mathrm{string}} \approx (8\pi f_a^2 H/\delta) \sqrt{\xi_*} \log_*$.

If the spectrum is exactly scale invariant at large $k$, such that $q = 1$, then $\delta \propto \log(m_r/H)$. Defining $\delta = \delta_1 \log(m_r/H)$ in this case we compute $\delta_1 = 6.2 \pm 0.4$. The axion number density from strings is then $n_a^{\mathrm{string}} \approx (8\pi f_a^2 H/\delta_1) \sqrt{\xi_*}$. At $1\sigma$ we find that $q$ could be as low as $q \approx 0.98$. For $q < 1$ the quantity $\delta$ increases for increasing UV cut-offs like $(m_r/H)^{1-q}$; in particular, for $q = 0.98$ and $\log m_r/H = 70$ we calculate $\delta = 820 \pm 50$. Thus, accounting

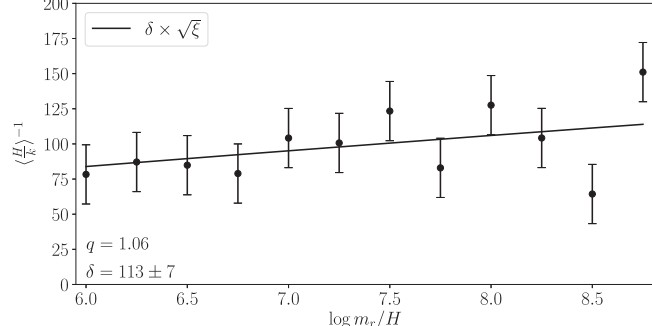

**Fig. 4 Evolution of the axion number density.** The inverse expectation value $\langle H/k \rangle^{-1}$ is computed using the instantaneous axion spectrum $F(k/H)$ by numerically integrating the spectrum to $k/H = x_{\max} = 50$ and then analytically integrating the power law distribution $F(x) \propto x^{-q}$ from $x_{\max}$ to the UV cut-off at $k/H \sim e^{\log_*}$ for $\log_* \approx 65$. The data are illustrated along with their 68% uncertainties. For $q > 1$ the expectation value does not strongly depend on the UV cut off but is instead a function of the effective IR cut-off, which is set by $\xi$ such that $\langle H/K \rangle^{-1} = \delta \sqrt{\xi}$ for some parameter $\delta$, which we determine by fitting this model to the numerical data as illustrated here. Smaller values of $\delta$ correspond to larger axion number densities and thus large axion DM densities. Here, we illustrate the result for the maximum allowed $q$ of 1.06, which leads to the smallest $\delta$ consistent with our simulation results.

for the uncertainty on $q$ from our simulations we find that $\delta$ is in the range $\delta \in (106, 870)$.

Let us more precisely define the time $t_*$ as the time when the axion field becomes dynamical, which is when $3H(t_*) = m_a(t_*)$, for a time-dependent mass $m_a(t)$ that is increasing rapidly during the QCD phase transition[30]. The axion string network is observed to collapse around $t_*$ (see, e.g. Buschmann et al.[26]), meaning that at times $t \gtrsim t_*$ axion number density is conserved. Assuming axion number density conservation allows us to relate the present-day DM abundance to the expression for $n_a^{string}$ at $t_*$ (see Methods section):

$$\Omega_a^{str} \approx 0.12\, h^{-2} \left( \frac{f_a}{4.3 \cdot 10^{10} \text{GeV}} \right)^{1.17} \frac{107}{\delta} \sqrt{\frac{\xi_*}{17}} \frac{\log_*}{70}. \quad (2)$$

Axions produced from domain wall and misalignment dynamics during the QCD phase transition provide a sub-dominant contribution to the DM density[26]: $\Omega_a^{QCD} \approx 0.017\, h^{-2} (f_a/4.3 \cdot 10^{10} \text{GeV})^{1.17}$. The DM abundance as measured by the Planck Observatory using the cosmic microwave background is $\Omega_{DM} = (0.12 \pm 0.0012)h^{-2}$, with $h$ the Hubble rate scaling factor[37]. Adding in the contribution from the QCD phase transition $\Omega_a^{QCD}$, and assuming $q \in (0.98, 1.06)$, we find that the $f_a$ that gives rise to the observed DM abundance should be in the range $f_a \in (3.1 \times 10^{10}, 1.4 \times 10^{11})$ GeV ($m_a \in (40, 180)\,\mu\text{eV}$), where for the lower $f_a$ bound we have conservatively allowed for the possibility that at $t_*$ the remaining energy density in strings is instantaneously deposited into axions with spectrum $F$, raising the string-induced DM density by a factor of $3/2$, though in actuality this contribution is likely smaller since the spectrum shifts towards the UV as $m_a(t)$ increases. If the index is scale invariant ($q = 1$), then we predict $m_a = 65 \pm 6\,\mu\text{eV}$.

## Discussion

In this work, we provide the largest and highest-resolution simulation of the axion string network to-date by making use of an AMR framework that allows us to resolve the axion string cores while maintaining lower resolution over the majority of the simulation volume. Our AMR approach may be used in the future to simulate the axion dynamics at the QCD epoch where domain walls form and the string network collapses[26] and to study axion-

like particle string networks that produce gravitational wave radiation[36,38,39].

Our results have important implications for axion direct detection experiments, as our preferred mass range of $(40, 180)\,\mu\text{eV}$ is higher than that which may be probed by two of the main dedicated experiments that are aiming to test this cosmological scenario, *ADMX*[40] and *HAYSTAC*[41]. On the other hand, this mass range may be probed by *ADMX* with future searches[42], by the *MADMAX* experiment[43,44], and by the proposed plasma haloscope[45]. Our work motivates focusing experimental efforts on this mass range. The dominant source of uncertainty on $m_a$ in our estimates arises from the index $q$, which we find does not evolve with $\log m_r/H$ and is in the range $(0.98, 1.06)$; this range is statistics-limited and will shrink with future simulation efforts using AMR, leading to more precise predictions that can in turn better inform experimental efforts.

## Methods

**Simulation framework.** We decompose the complex PQ scalar field as $\Phi = (\phi_1 + i\phi_2)/\sqrt{2}$ and assume a radiation-dominated cosmological background. In this notation the axion field is given by $a(x) = f_a \arctan2(\phi_2, \phi_1)$ and the radial mode by $r(x) = \sqrt{\phi_1^2 + \phi_2^2} - f_a$. The EOM can be derived from the Lagragian in (1) and expressed in the dimensionless fields $\psi = \phi/f_a$ as

$$\psi_1'' + \frac{2}{\eta}\psi_1' - \bar{\nabla}^2\psi_1 + \lambda\psi_1\left[\eta^2\left(\psi_1^2 + \psi_2^2 - 1\right) + \frac{T_1^2}{3f_a^2}\right] = 0, \quad (3)$$

$$\psi_2'' + \frac{2}{\eta}\psi_2' - \bar{\nabla}^2\psi_2 + \lambda\psi_2\left[\eta^2\left(\psi_1^2 + \psi_2^2 - 1\right) + \frac{T_1^2}{3f_a^2}\right] = 0, \quad (4)$$

with $T_1$ defined as the temperature when $H(T_1) = f_a$. Here, primes denote derivatives with respect to $\eta$ while the spatial gradient $\bar{\nabla}$ is taken with respect to $\bar{x} = R_1 H_1 x$. We chose $\lambda = 1$ without loss of generality and the ratio $(T_1/f_a)^2$ is given by

$$\left( \frac{T_1}{f_a} \right)^2 \approx 8.4 \times 10^5 \left( \frac{10^{12}\,\text{GeV}}{f_a} \right). \quad (5)$$

Note that the PQ breaking scale $f_a$ is degenerate with the choice of physical box size $L$ and dynamical range in $\eta$. This implies that one has to perform only a single simulation, which can be reinterpreted through trivial rescaling for different axion masses.

Using an AMR technique means that some parts of our simulation volume are run at a higher spatial (and temporal) resolution than other parts. Our implementation is based on AMReX[31], a software framework for block-structured AMR.

Our simulation starts out with a uniform grid of $N_0 = 2048^3$ cells, which we will refer to as the coarse level. We generate thermal initial conditions with wavenumber up to 25 in each spatial direction at an initial time $\eta_i = 0.1$. See Buschmann et al.[26] for details of how the initial state for $\Phi$ is generated from the thermal correlation functions. The comoving box length of our simulation volume is $L = 120$ with periodic boundary conditions. This implies the simulation contains $(120)^3$ Hubble volumes at $\eta = 1$. Our starting time is $\eta = 0.1$. Note that the comoving spatial difference $\Delta x_0 = L/N_0$ between lattice points is such that our initial state for $\psi$ is smooth during the initial stages of the PQ phase transition (i.e., the structure in $\psi$ is resolved by multiple grid sites).

Linearization of the EOM in (3) and (4) reveals that the system of PDEs is strongly hyperbolic and admits stable evolution with Runge-Kutta time integration and the method of lines (MoL). We chose to reduce the EOM to first order in time by defining the conjugate momentum $\Pi_{1,2} \equiv \psi'$ and evolving $\Pi_{1,2}$ and $\psi_{1,2}$ independently. The EOM in (3) and (4) is solved using the strong-stability preserving Runge-Kutta (SSPRK3) method. This method is of third-order and as such one higher than the often used leapfrog integration scheme. We find that this method provides the best trade-off between numerical stability and computational costs including memory consumption when compared against a second- and fourth-order Runge-Kutta method. At the coarse level, the time step is $\Delta\eta_0 = 0.02$, satisfying the Courant-Friedrichs-Lewy (CFL) condition at $\Delta\eta_0/\Delta x_0 \approx 1/3$. The laplacian in the EOM is computed to sufficient accuracy by a second-order finite difference method.

A grid of $N_0 = 2048^3$ cells will not be able to resolve string cores at late times. To maintain resolution we periodically refine a volume around strings, which means decreasing the grid spacing by a factor of 2 in a local volume (see Fig. 1). We refer to the volumes with different resolutions as levels $\ell$ with the coarse level being level $\ell = 0$. Each level differs from each other not only in spatial resolution,

$\Delta x_\ell = \Delta x_0/2^\ell$, but also in temporal resolution to locally satisfy the CFL condition, $\Delta\eta_\ell = \Delta\eta_0/2^\ell$. The higher-resolution lattice on level $\ell$ is determined by fourth-order spatial interpolation of the coarser level $\ell-1$ if no data at that location and level exists. Since different grid spacing and time step sizes are used simultaneously, each level is evolved independently and then synchronized appropriately. This is known as the subcycling-in-time approach and requires fourth-order spatial interpolation and second-order temporal interpolation during synchronization. The simulation is insensitive to the exact order of the interpolation used. See the AMReX documentation[31] for more information about the technical details of the AMR approach.

We add an additional level each time the string width $\Gamma$ drops below four grid sites at the current finest level, i.e. at $\eta \approx 3, 6, 12,$ and $24$ ($\log m_r/H \approx 2.6, 3.9, 5.3, 6.7$), leading to a total number of 5 levels. A 6th level is added at $\eta \approx 64(\log m_r/H \approx 8.7)$ when the string width drops to $3\Delta x_{\ell=4}$. See Supplementary Fig. 1 for an illustration of the respective string core resolution at different times in our simulation, compared to the resolution achieved in the static lattice simulation in Gorghetto et al.[27]. Note that to match the resolution of the finest level on a uniform grid would require a stunning $65,536^3$ cells.

We use a tagging algorithm to decide on which local volumes to refine, with cells tagged ensured to be within a refined volume. In total we use three different tagging criteria that target (i) string cores, (ii) large gradients in $\psi$, and (iii) short wave-length radiation emitted by strings:

- String cores are identified using the procedure described in Fleury and Moore[33] (Appendix A.2). This involves finding plaquettes that are being pierced by strings. The cell at the low-index corner of a pierced plaquette is tagged.
- As strings decay the resulting radiation can produce large gradients in the field. To ensure sufficient resolution we tag cells with $\Delta x_\ell^2 \nabla^2 \psi_{1,2} > 0.04$. The precise numerical value is of phenomenological origin and has proven to work well in our simulation setup.
- String radiation into radial modes is highly suppressed at late times yet it can cause numerical instabilities if not sufficiently resolved. To avoid a numerical breakdown we tag cells at the coarse level where $\Delta x_0\eta^3\nabla^2(\psi_1^2 + \psi_2^2) > 4$ is fulfilled.

Certain string dynamics, in particular strings that are reconnecting, can cause large shock waves that travel away from strings. The typical scale of these wave fronts is related to the string width and they therefore requires good spatial resolution as well. The refinement criteria (ii) and (iii) track these wave fronts in the two field components as they would not be otherwise captured by the first refinement criteria. Strings are not stationary and thus the grid layout has to be adjusted periodically. As this is computationally expensive we re-grid level $\ell$ only every $\Delta\eta_\ell = 0.2/2^\ell$. However, we ensure that within this time interval even the fastest moving strings with $v = c$ are always at least a full string width away from any coarse-fine boundary. This is done by leaving a buffer zone of 11 grid sites around each tagged cell that is refined as well.

The simulation was performed on NERSC's Cori XC40 system using 1024 KNL nodes (Intel Xeon Phi Processor 7250) with, in total, 69,632 physical CPU cores and over 98 TB DDR4 RAM. It ran for about 74 h (~5.2 Million CPU hours) in a hybrid OpenMP/MPI mode. Some of the smaller systematic tests ran on NERSC's Perlmutter system with up to 128 NVIDIA A100 GPUs.

**The string length per Hubble $\xi$.** We compute the string length per Hubble $\xi$, defined in the main Article, using the algorithm from Fleury and Moore[33] that involves counting string-pierced plaquettes; our measured values for $\xi$ are illustrated in Fig. 2. We then fit the model

$$\tilde{\xi} = \frac{c_{-2}}{\log^2 m_r/H} + \frac{c_{-1}}{\log m_r/H} + c_0 + c_1 \log m_r/H \qquad (6)$$

to this data, though this fit is made complicated by the fact that it is difficult to estimate statistical uncertainties from our $\xi$ measurements. We thus determine these uncertainties in a data-driven way. Given that we expect the uncertainties to be statistical in nature, and thus determined by the finite simulation volume, we assign uncertainties to each measurement such that the uncertainty at a given $\log_i \equiv \log m_r/H(t_i)$ value is $\sigma_{\xi_i} = \sigma_0 e^{-3\log_i/4}$. Here, the factor $e^{-3\log_i/4}$ is the time-dependence of the square-root of the number of Hubble patches per simulation box, which is a proxy for the square-root of the number of independent string segments in the simulation volume. We then treat $\sigma_0$ as a nuisance parameter that we profile over during the fit. In particular, the likelihood is

$$\mathcal{L}_\xi[\xi; \mathcal{M}_\xi, \{c, \sigma_0\}] = \prod_i \frac{\exp\left[-\frac{(\xi_i - \tilde{\xi}_i)^2}{2\sigma_{\xi_i}^2}\right]}{\sqrt{2\pi\sigma_{\xi_i}^2}}, \qquad (7)$$

with $\xi_i$ and $\tilde{\xi}_i$ denoting the data and model prediction, respectively, at the time labeled by $\log_i$. Note that we denote the model by $\mathcal{M}_\xi$ with model parameter vector $c = \{c_{-2}, c_{-1}, c_0, c_1\}$ in addition to $\sigma_0$. The uncertainties in Fig. 2 arise from the best-fit $\sigma_0$.

**Construction of the axion energy density spectrum.** In order to compute the axion energy density spectrum, we consider the screened time-derivative of the axion field, which is defined by

$$\dot{a}_{\mathrm{scr}}(x) = f(x)\,\dot{a}(x). \qquad (8)$$

In this definition, we include a function $f$ that screens out the locations of strings, which appear as discontinuities in the axion field and its derivative. We consider three choices of the screening function:

$$f(x) = \left[1 + r(x)/f_a\right]^2 \qquad (9)$$

$$f(x) = 1 + r(x)/f_a \qquad (10)$$

$$f(x) = 1 \quad \text{(no mask)}. \qquad (11)$$

In this work our fiducial results use (9) such that $\dot{a}_{\mathrm{scr}}(x) = \psi_1(x)\dot{\psi}_2(x) - \dot{\psi}_1(x)\psi_2(x)$. The screening in (10) reproduces that of[27] while (11) corresponds to no string screening. Because $1 + r(x)/f_a \approx 1$ at locations far away from string cores, screening as in (9) and (10) only modify the axion time derivative in the direct vicinity of strings. As shown in Supplementary Fig. 7, the results presented in this work are relatively insensitive to the choice of screening function, which can be understood from the fact that we study the emission at spatial scales well beyond the string width.

The axion energy density spectrum within our simulation can then be computed as in Gorghetto et al.[27] by

$$\frac{\partial\rho_a}{\partial k} = \frac{|k|^2}{(2\pi L)^3} \int d\Omega_k |\tilde{\dot{a}}_{\mathrm{scr}}(k)|^2, \qquad (12)$$

where $\tilde{\dot{a}}_{\mathrm{scr}}(k)$ is the Fourier transform of the field $\dot{a}_{\mathrm{scr}}$. We compute this energy density spectrum with the HACC SWFFT algorithm[46] applied to the axion time derivative computed at the coarsest level of spatial resolution. After we have compasuted $d\rho_a/dk$ using the fast Fourier transform (FFT), we then bin our FFT data in 1774 equal-sized bins between $k = 0$ and the maximum $k$, corresponding to $k_{\mathrm{com}}^{\mathrm{max}}/2\pi = 1024\sqrt{3}/L$. This binned spectrum is then used in our subsequent analysis.

**Measuring the string tension.** We compute the effective string tension realized in our simulation following the procedure described in refs. [24,27]. We first compute the average energy density within our entire simulation volume using

$$\rho_{\mathrm{tot}} = \langle|\partial\Phi|^2 + \lambda\left(|\Phi|^2 - \frac{f_a^2}{2}\right)^2\rangle. \qquad (13)$$

We then compute the average axion and radial mode energy densities by

$$\rho_a \approx \langle\dot{a}^2\rangle, \qquad (14)$$

$$\rho_r \approx \langle\frac{1}{2}\dot{r}^2 + \frac{1}{2}(\nabla r)^2 + \frac{\lambda}{4}(r^2 + 2rf_a)^2\rangle. \qquad (15)$$

In computing $\rho_a$ and $\rho_r$, we mask regions of the simulation volume that are at the highest level of refinement to exclude string contributions. Note that in computing both $\rho_{\mathrm{tot}}$ and $\rho_r$, we neglect the small contribution of the thermal mass in (1). The string energy density is then straightforwardly obtained from

$$\rho_s = \rho_{\mathrm{tot}} - \rho_a - \rho_r. \qquad (16)$$

Using the string energy density, we may determine the effective tension by

$$\mu_{\mathrm{data}} = t^2 \rho_s/\xi, \qquad (17)$$

with the subscript "data" denoting the measured value, which can be compared to the theoretically expected string tension at large values of $\log m_r/H$:

$$\mu_{\mathrm{th}} \approx \pi f_a^2 \log\frac{m_r}{H}. \qquad (18)$$

This comparison is illustrated in Supplementary Fig. 2 for times between $\log m_r/H = 8$ and $\log m_r/H = 9$.

Importantly, we only want to compare the leading log behavior of $\mu_{\mathrm{data}}$ and $\mu_{\mathrm{th}}$. Moreover, the addition of a refinement level at $\log m_r/H \approx 8.7$ changes the effective UV cutoff in the numerical calculation, leading to a discontinuity in the measured effective tension. To analyze the effective tension, we thus adopt a simple logarithmic growth model for the effective tension

$$\mu = \begin{cases} \mu_1 f_a^2 \log m_r/H + \mu_b, & \log m_r/H \le 8.7 \\ \mu_1 f_a^2 \log m_r/H + \mu_a, & \text{else}, \end{cases} \qquad (19)$$

which allows for a different constant offset before ($\mu_b$) and after ($\mu_a$) the addition of the refinement level but enforces uniform logarithmic growth of the string tension. We use a Gaussian likelihood with data-driven uncertainty on the $\mu_{\mathrm{data}}$ values $\sigma_\mu$; we treat $\sigma_\mu$ as a nuisance parameter in addition to $\mu_{a,b}$. Profiling over the nuisance parameters we determine $\mu_1 = 3.7 \pm 0.5$, which should be compared to the theoretically expected value $\mu_1 = \pi$.

Given that we compute the effective string tension from the time-evolution of the total energy density, as highlighted in Eq. (16), the agreement between the measured tension and the theoretical prediction may be seen as a test that our numerical procedure does not have a source of numerically induced energy leakage, at least not at the level of precision that we may measure in this test.

**Instantaneous emission analysis.** Here we describe the method by which we fit a power-law model to the instantaneous emission spectrum. Up to an overall normalization, the instantaneous emission spectrum is given by

$$F\left(\frac{k}{H}\right) \propto \frac{1}{R^3} \frac{\partial}{\partial t}\left(R^3 \frac{\partial \rho_a}{\partial k}\right). \quad (20)$$

In our simulation framework, time evolution is performed in terms of $\eta$ and hence the instantaneous emission $F_i$ at conformal time $\eta_i$ is calculated by numerical finite difference as

$$F_i\left(\frac{k}{H}\right) \propto \frac{1}{\eta_i^4}\left(\frac{\eta_{i+1}^3 \frac{\partial \rho_{i+1}}{\partial k} - \eta_i^3 \frac{\partial \rho_i}{\partial k}}{\eta_{i+1} - \eta_i}\right), \quad (21)$$

where $\partial \rho_i/\partial k$ is the axion energy density spectrum at $\eta_i$. At each $\eta_i$, we consider a power-law model of the form

$$f\left(\frac{k}{H}; \{A, q\}\right) = A\left(\frac{k}{H}\right)^{-q} \quad (22)$$

and adopt the parametrized form

$$\sigma\left(\frac{k}{H}; \{B, p, C\}\right) = B\left(\frac{k}{H}\right)^{-p} + C \quad (23)$$

to describe the combined statistical and systematic uncertainty in the data. We then analyze the data at each $\eta_i$ with the Gaussian likelihood $\mathcal{L}_i$, which is of the form

$$\mathcal{L}_i[\mathbf{d}_i; \mathcal{M}_i] = \prod_j \frac{1}{\sqrt{2\pi}\sigma_j} \exp\left[-\frac{1}{2}\left(\frac{d_{i,j} - f_j}{\sigma_j}\right)^2\right] \quad (24)$$

where $d_{i,j}$ is the value of the numerically computed instantaneous emission spectrum at the $j$th value of $k/H$ computed at time $\eta_i$. The model predictions for the mean and the error at the $j^{\text{th}}$ value of $k/H$ are specified by the model parameters $\mathcal{M}_i = \{A_i, q_i, B_i, p_i, C_i\}$ for each time $\eta_i$. The values of $k/H$ and associated data that enter the likelihood are restricted to satisfy $k/H > x_{\text{IR}}$ and $k/H < x_{\text{UV}}^{-1} m_r/H$.

In performing the analysis, we only analyze emission spectra which contain at least 10 bins between $k_{\text{IR}} \equiv H x_{\text{IR}}$ and $k_{\text{UV}} \equiv m_r/x_{\text{UV}}$. We make the fiducial analysis choices of using the screening function of Eq. (9), $k_{\text{IR}} = 50H$ and $k_{\text{UV}} = m_r/16$, and using a finite difference in time-spacings corresponding to $\Delta \log m_r/H = 0.25$. The impact of varying these fiducial choices, which is marginal, is illustrated in the Supplementary Figs. 4–7.

Using the likelihood in Eq. (24), we determine the maximum likelihood estimate $\hat{q}_i$ for the emission index at each $\eta_i$. Since the likelihoods are quadratic to very good approximation, we also determine Gaussian uncertainties $\sigma_{q_i}$ on $\hat{q}_i$ at each $\eta_i$ by $1/\sigma_{q_i}^2 = -\partial^2/\partial_{q_i}^2 \log \mathcal{L}_i$ evaluated at the likelihood-maximizing model parameters. After obtaining $\hat{q}_i$ and $\sigma_{q_i}$ at each $\eta_i$, we join the results to study the possible evolution of $q$. We use a Gaussian likelihood

$$\mathcal{L}_q[\mathbf{q}; \mathcal{M}_q, \sigma] = \prod_i \frac{\exp\left[-\frac{(\hat{q}_i - \tilde{q}_i)^2}{2(\sigma^2 + \sigma_{\hat{q}_i}^2)}\right]}{\sqrt{2\pi(\sigma^2 + \sigma_{\hat{q}_i}^2)}} \quad (25)$$

where $\tilde{q}_i$ is the model prediction at time $\eta_i$ specified by parameters $\mathcal{M}_q$. We include an additional error term $\sigma$ as a nuisance parameter which is added in quadrature with the data-driven $\sigma_{q_i}$ to address possible systematic effects. In this work, we consider two possibilities for the evolution of $q$, the first that $q$ grows linearly as $q(\log m_r/H) = q_1(\log m_r/H) + q_0$ and the second that $q$ is constant such that $q(\log m_r/H) = c_0$. As in our analysis of the individual instantaneous emission spectra, the maximum likelihood estimates and uncertainties of the parameters $\sigma$, $q_0$, and $q_1$ can be determined via standard frequentist techniques.

**AMR convergence.** We perform two different, smaller-scale simulations to test the convergence of the AMR technique. Both simulations use a comoving side length of $L = 23/(R_1 H_1)$, which allows us to simulate to a final state at $\log \sim 5.8$ before running into finite-volume effects. One simulation uses a static grid with $2048^3$ sites, which is large enough that the string width is still resolved by four sites at the final state. This simulation serves as the baseline, as it over-resolves the entire field at all times. The second simulation uses AMR with a coarse level resolution of $512^3$ sites. By adding two additional refinement levels at $\log \sim 3.1$ and $\log \sim 4.5$ we ensure that strings are resolved by at least four grid sites at all times, though the resolution may be lower away from the strings. The initial states for both simulations are identical; that is, the $512^3$ initial state is a down-sampled version of the $2048^3$ initial state. All other parameters are identical to that of our main simulation. Note that the static-grid simulation is achieved by running within AMReX but without employing any of its AMR capabilities.

For both simulations we compute our quantities of interest, the string length per Hubble volume $\xi$ and the instantaneous axion radiation spectrum. In Supplementary Fig. 11 we present the relative difference in string length between the two simulations. At all times the relative difference between both simulations is <0.4% and, more importantly, the relative difference is centered around zero with no observable drift in either direction. In Supplementary Fig. 12, we compare the relative difference in emission spectra obtained between the two simulations, where the differences are also at the sub-percent level for $k \lesssim m_r/4$, which is the largest $k$ considered in this work. Note that these spectra are computed by comparing the states between $\log m_r/H = 4.953$ and $\log m_R/H = 5.790$. The FFT for the static grid is computed from the $2048^3$ state, while for the AMR simulation the FFT is computed from the coarse level ($512^3$); thus, some small differences may be expected. This is because of the fact that down-binning to the coarse level is equivalent to an effective low-pass filter, which suppresses the power of the high-$k$ modes and thus leads to a small spectral distortion. However, in both this test and in our fiducial analysis we calculate that this spectral distortion on the index is at the one percent level or less and thus subdominant compared to our statistical uncertainty. While this test did not evolve to high enough log to fit for the index $q$, since such evolution would be challenging with a static grid, we can infer from the differences in the spectra that the differences in $q$ would likely also be at the sub-percent level and thus subdominant compared to our statistical uncertainty on the final $q$ measurement, which is at the level of ~5%.

**Dependence on the initial state.** We perform a set of simulations to test the dependence of our results on the initial state. In particular, in our fiducial simulation we were forced to make a choice for the maximum wavenumber for the modes included in our initial state. In this section we provide evidence that our results are not sensitive to that choice by performing multiple, smaller simulations. Each of the smaller simulations uses a $L = 44/(R_1 H_1)$ box length and a coarse level resolution of $1024^3$ sites. With three extra refinement levels this allows us to simulate to $\log \sim 7$. We perform two sets of simulations with a maximum wavenumber of 5 and 25. For each set we run two different initial state realizations to enhance statistics. The physical size of the highest-$k$ mode can be characterized by $L/k_{\text{max}}$, with our main simulation corresponding to $L/k_{\text{max}} = 120/25 = 4.8$. Thus, we run additional simulations with $L/k_{\text{max}} \sim 8.8$ and 1.8 to study the effects of minimum physical wavelengths much smaller and much larger than in our fiducial simulation.

For each simulation we compute the string length per Hubble volume $\xi$ and the instantaneous emission spectrum $F(k/H)$. In Supplementary Fig. 13 we compare the evolution of $\xi$ from our systematic tests with that of our main simulation. In this section, and this section only, we determine the error on $\xi$ by $\sigma_\xi = \xi \sigma_{N_{\text{seg}}}/N_{\text{seg}}$, where $\sigma_{N_{\text{seg}}}$ is the Poissonian error on the independent number of string elements $N_{\text{seg}} = \ell_{\text{tot}}/L_{\text{corr}} = 4\xi^{3/2}(L/\eta)^3$. Here, $\ell_{\text{tot}} = V\xi/t^2$ is the total number of string segments in a volume $V$ with a correlation length of $L_{\text{corr}} = 1/(H\sqrt{\xi})$. The simulation volumes of both initial state realizations are combined in this procedure. Note that we determine the uncertainties this way in order to avoid having to perform fits of functional forms to the low $\log \xi$ data. While the field configurations of the three $L/k_{\text{max}}$ variations start out drastically different, the differences appear to diminish at later times. This supports the hypothesis of Gorghetto et al.[39], which is that the axion string evolution approaches an attractor solution.

In Supplementary Fig. 14, we stack the emission spectra from the two systematic variations and their best-fit power-law model, using the final-state data at $\log \sim 7$. We compare these results to that from our fiducial simulation. The systematic variations and the fiducial simulation demonstrate mutual compatibility at the expected statistical precision, as indicated in the right panel of Supplementary Fig. 14. Note that for these fits, to ensure a suitably large analysis region, we choose an IR cutoff of $30H$ and a UV cutoff of $m_r/8$ rather than our fiducial analysis choice of $50H$ and $m_r/16$. Results for the individual simulations without stacking are presented in Supplementary Tab. V.

**DM abundance calculation.** Here we describe the calculation of the DM abundance from the quantity $n_a^{\text{string}}$, which is described in the main Article. Define $\Lambda \equiv 400$ MeV; then the temperature-dependent axion mass is well characterized by a power-law[47]:

$$m_a^2(T) = \frac{\alpha_a \Lambda^4}{f_a^2 (T/\Lambda)^n}, \qquad T \gg \Lambda, \quad (26)$$

with $\alpha_a$ and $n$ dimensionless constants. The most recent lattice simulations agree with the dilute instanton gas approximation and support $\alpha_a = (4.6 \pm 0.9) \times 10^{-7}$ for $n \approx 8.16$[48], which are the values we assume in this work (note that these uncertainties are sub-dominant to those from the axion production from strings from our simulations). We also approximate the temperature-dependent number of relativistic degrees of freedom as $g_*(T) \approx g_*^0(T/\text{MeV})^\gamma$, with $g_*^0 \approx 50.8$ and $\gamma \approx 0.053$, which has been shown to match the full result for $g_*(T)$ up to a few percent over the temperature range $800 < T < 1800$ MeV relevant for this calculation[49]. We also assume that the numbers of radiation and entropy degrees of freedom are the same over the temperature range of interest, since the difference between these is also at the level of a few percent[48] and thus a sub-dominant source of uncertainty.

The temperature $T_*$ is defined as the temperature at which $3H(T_*) = m_a(T_*)$; using $H(T_*) = \pi\sqrt{g_*(T)/90}T_*^2/M_{pl}$, with $M_{pl}$ the reduced Planck mass, we may solve explicitly for $T_*$. We assume that the string network evolves uninterrupted up to $T_*$ but that for $T < T_*$ it quickly evaporates and is not a significance source of axions (but see below). In this approximation axion number density is conserved for $T < T_*$, so that we may write the axion DM abundance today as in Eq. (2). Note that $\Omega_a^{str} \propto f_a^{(6+n+\gamma)/(4+n+\gamma)} \approx f_a^{1.17}$, which is the same scaling as for $\Omega_a^{QCD}$, since for both contributions the $f_a$-scaling has the same origin (see e.g. Buschmann et al.[26]).

While we do not simulate the QCD phase transition in this work, it is important to keep in mind that the string network does evolve non-trivially during the QCD epoch. As illustrated in the simulations in Buschmann et al.[26], the string network collapses rapidly after $T_*$. In particular, the string network in Buschmann et al.[26] was completely gone at temperatures of order $T \sim T_*/1.5$. Note that we and Buschmann et al.[26] assume a domain wall number $N_{DW}$ of unity so that domain walls are unstable. For an alternative scenario with $N_{DW} > 1$ where domain walls can exist well beyond the QCD phase transition see e.g. Hiramatsu et al.[50]. In our approximation where the string network evolves uninterrupted until $T_*$ the string network has energy $\rho_s = 4H^2(T_*)\mu(T_*)\xi_*$ at $T_*$. Between $T_*$ and $\sim 1/1.5T_*$ all of that energy is transferred to axion radiation. However, it is likely that the spectrum of radiation during this collapse is shifted to the UV compared to the function $F(k/H)$ from before the mass turns on, since after $T_*$ the axion mass $m_a(T)$ is much larger than Hubble and thus provides an IR cut-off for the radiation spectrum that is further in the UV compared to that for the axion-string network prior to the QCD phase transition. Since the spectrum is shifted towards the UV, it should produce less axions by number density and thus be less important for the final DM abundance. Still, in order to be conservative we estimate the maximum amount of DM that may be produced by the string network by assuming that at $T_*$ all of the energy density in $\rho_s$ is transferred instantaneously to axions with spectrum $F(k/H)$. This provides a contribution to the axion energy density $\Omega_{str}^{decay} \approx \Omega_a^{str}/2$, with $\Omega_a^{str}$ being the contribution to the DM abundance from axions produced prior to $T_*$. We allow for this possibility when determining that the maximum allowed axion mass is 180 μeV, but we do not include this contribution when estimating the minimum allowed axion mass of 40 μeV.

In this work we assume that the radial mass, $m_r$, is of order the decay constant $f_a \sim 10^{10} - 10^{12}$ GeV. However, one possibility is that $m_r \ll f_a$, as may happen in e.g. supersymmetric theories where $m_r$ is related to the supersymmetry breaking scale; in this case, $m_r \gtrsim$ TeV is possible[51]. If $m_a \sim$ TeV, then $\log(m_r/H) \sim 50$, which is large enough such that our conclusion that $m_a \in (40, 180)$μeV produces the correct DM abundance is still valid in this scenario.

Note that in these estimates we must perform the fit of the model $\delta \times \sqrt{\xi}$ to the $\langle H/k \rangle^{-1}$ data illustrated in Fig. 4. The $\langle H/k \rangle^{-1}$ data do not have easily estimated uncertainties and so, as we have illustrated multiple times already, we determine these uncertainties in a data-driven way by assigning the uncertainties of all data points a value $\sigma$, which we profile over when determining $\delta$. The uncertainties in Fig. 4 reflect the best-fit value of $\sigma$.

Lastly, the derivation above assumes that number-changing processes are not important in the QCD phase transition since $|a/f_{al}| \lesssim 1$. Note that the formula in the main Article for $\langle (a/f_a)^2 \rangle$ for the string-induced axion radiation arises from the relation $\langle (a/f_a)^2 \rangle = (1/f_a^2) \int dk\, d\rho_a/dk (1/k^2)$, with $d\rho_a/dk = \int^t dt' \Gamma'/H'(R'/R)^3 F(kR/R'H')$ and primes denoting quantities evaluated at $t'$.

**Semi-analytic analysis of string evolution**. In the main Article we pointed to an argument related to the logarithmically increasing string tension for why $\xi$ may be expected to increase logarithmically in time as well. Here, we expand upon that argument as well as give an argument for why $q = 1$ may be expected for the spectrum. As the string network evolves in the scaling regime axions are produced at a rate $\Gamma_a \approx 2H\rho_s$, where $\rho_s = \xi\mu/t^2$ is the energy density in strings, and $\mu \approx \mu_0 \log(m_r/H)$ is the string tension, to leading order in large log. Recall that $\mu_0 = \pi f_a^2$. The tension $\mu$ has a logarithmic divergence that is regulated in the IR by the scale of string curvature $\sim H^{-1}$ because of energy associated with the axion field configuration, which wraps around the string. Physically we may imagine that the long strings are composed of a random walk of smaller segments that we refer to as correlation lengths, which may evolve dynamically and straighten on timescales of order $H^{-1}$. Denote the number density of correlation lengths as $n_c$. Then, we may relate $\Gamma_a = n_c dE_c/dt$, where $dE_c/dt$ is the power transferred to axions by the straightening correlation lengths. Previous studies of collapsing closed string loops and straightening string kinks have shown that the loops and kinks lose energy as $dE/dt = -\alpha f_a^2$, with $\alpha \sim \mathcal{O}(1-10)$, regardless of the loop and kink sizes[14,20,52,53]. We assume that the correlation lengths radiate as $dE_c/dt = -\alpha f_a^2$ for some $\alpha \sim 1 - 100$. Solving the energy balance equation then leads to a time-dependent correlation length $L_c \approx \alpha/[\pi H \log(m_r/H)]$ for large log. Let us now assume that there are $\sim N_{str}$ strings in total per Hubble patch, with each string composed of a random walk of smaller correlation lengths. This then implies that at large $\log(m_r/H)$ the parameter $\xi$ scales as $\xi \approx N_{str}(\pi/4\alpha)\log(m_r/H)$, which reproduces the observed scaling for $N_{str} \sim$ few, consistent with the simulation data as illustrated in Fig. 2, and $\alpha \sim \mathcal{O}(10)$.

One of the most important results of this Article is the result that $q \approx 1$, to within $\sim 5\%$. In order to further support this result, we show visually that the string

distribution is approaching an attractive solution that supports $q = 1$. String loops can be characterized by the parameter $n_\ell$, which is the number of string loops with size smaller than $\ell$ at a time $t$, as well as by $\xi_\ell$, which is the total length of string loops with size smaller than $\ell$. In Supplementary Fig. 3 we illustrate $\xi_\ell/\xi_\infty$ versus the length $\ell$ at various times, with $\xi_\infty = \xi$. Visually, it is clear that as time progresses, the string loop distribution approaches an attractor solution, whose validity is extending over an increasingly large range of lengths. This sort of attractor solution for the loop distribution was also found for the fat string approximation in Gorghetto et al.[24] but here we are able to show that this also holds in the physical case. Given the importance of this distribution, we numerically fit a power law model to the data using the same procedure described in Methods Sec. I. The treatment of uncertainties and definition of the Gaussian likelihood is analogous to that used for the instantaneous emission spectrum, with a power-law model of the form $\xi_\ell = D\ell^m$. We perform the fit at various times $\eta_i$ to obtain corresponding indices $m_i$. The fitting range is $H\ell/\pi \in (8H/m_r, 1/2)$ to ensure we are within the attractor regime. We only include string loop distributions with at least 8 data points within the fitting range and $\log m_r/H \geq 4$. The results for $m_i$ are then joint using a Gaussian likelihood identical to that for $q$ in (25) assuming $m$ is time-independent. We find $m = 0.97 \pm 0.03$ with the fit illustrated in Supplementary Fig. 3.

Let us now show that $m = 1$ implies $q = 1$. From a $m = 1$ length distribution, we can calculate the number of strings loops with length between $\ell$ and $\ell + d\ell$ to be

$$\frac{d\xi_\ell}{d\ell} = \ell\frac{dn_\ell}{d\ell} = D,\tag{27}$$

for some constant $D$. We can determine the constant $D$ by using

$$\rho = \int d\ell\frac{d\rho}{d\ell} = \int d\ell\mu\ell\frac{dn_\ell}{d\ell} = D\mu\ell_{max} = \frac{\xi_{sub\,H}\mu}{t^2},\tag{28}$$

leading to $D \approx \xi_{sub\,H}/t^3$ with $\xi_{sub\,H}$ representing the total string length in sub-horizon sized string loops.

We are interested in the spectrum of axions emitted by the string network. A string loop of length $\ell$ emits axions dominantly at the fundamental frequency $k \sim 1/\ell$. Meanwhile, the string loop radiates energy at a rate $\frac{dE}{dt} = -\alpha f_a^2$. We can now combine all of this knowledge with Eq. (27) to find

$$F[k/H] \propto \alpha f_a^2\frac{dn_\ell}{dk} = \frac{\alpha f_a^2}{k}.\tag{29}$$

We thus find that given the attractive behavior seen in Supplementary Fig. 3, that the instantaneous spectrum of axions emitted by the network should be approaching $q = 1$. As a side-note, given this understanding of the string loop distribution, we can easily derive the energy density and spectrum of gravity waves emitted by string loops using $dE_{GW}/dt = -\alpha_{GW}G\mu^2$[39]. Note, however, that Gorghetto et al.[39] finds $q \approx 2$ for gravitational waves. A clue for reconciling these results may be found in the curvature distribution of the infinitely long straight strings. Taking the radius of curvature to be $\sim \ell$, the curvature distribution of the infinite string is approximately $\frac{d\xi_\ell}{d\ell} \sim \ell$ (see the discussion in the following paragraphs for a possible explanation for this curvature distribution). Applying the arguments in this section, we find that the infinitely long string radiates gravity waves with a spectrum of $q \approx 2$. If axions were dominantly radiated from the string loops while gravity waves were dominantly radiated from the infinitely long strings, the different spectra would be reconciled. It would be interesting to study this difference in more detail.

Finally, we conclude by giving analytic arguments for why Supplementary Fig. 3 takes the form that it does. Namely that at larger lengths, $\xi_\ell \propto \ell$, and at smaller lengths $\xi_\ell \propto \ell^2$. At small lengths, the string loops shrink due to the emission of axions giving $\ell(t) = \ell_0 - \alpha f_a^2(t - t_0)/\mu$ with $\ell_0$ being the initial loop size at a time $t_0$. If string loops are formed at a constant rate with a fixed length $\ell_0$, then $dn_\ell/d\ell \propto dn_{\ell_0}/dt =$ constant. Multiplying by $\ell$ and integrating, one finds that at small lengths $\xi_\ell \propto \ell^2$, in rough agreement with Supplementary Fig. 3.

Larger string loops shrink by intersecting the long, relatively straight, and infinite strings that carry most of the string length. The two strings will intersect at a rate $\Gamma_{int}$ given roughly by the average string speed over the average distance between strings. Upon intersecting the infinite string, the string loop loses a random amount of its string length. If the locations of the intersections are random, the probability distribution for the final length of the string loop, $\ell$, is proportional to its length. Thus an initial string loop of length $\ell_0$ has $dP/d\ell = 2\ell/\ell_0^2$. Putting this intuition into equation form, we find

$$\frac{dn_\ell}{dtd\ell} = -\Gamma_{int}\frac{dn_\ell}{d\ell} + \int_\ell^\infty d\ell_0\frac{dP}{d\ell}\frac{dn_{\ell_0}}{d\ell_0}\Gamma_{int}.\tag{30}$$

The first term on the right hand side gives the loss of loops due to intersections while the second term gives their production from larger loops of size $\ell_0$. Solving for the equilibrium distribution, we find that $dn_\ell/d\ell \propto 1/\ell$. As before, multiplying by $\ell$ and integrating, one finds that $\xi_\ell \propto \ell$ giving $q = 1$.

## Data availability

Due to the large size of the simulation output (>50 Terabytes), data products from this work are not stored in a public repository but may be made available by the

corresponding authors upon request. Supplementary animations are available at https://bit.ly/amr_axion and can be downloaded at Buschmann et al.[54].

## Code availability

The `AMReX` code framework used in this work is publicly available at https://amrex-codes.github.io/. Additional code may be made available upon request, though it is highly tailored for NERSC's Cori XC40 system.

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

## Acknowledgements

We thank Marco Gorghetto, Edward Hardy, David Marsh, Javier Redondo, Pierre Sikivie, Ofri Telem, Alejandro Vaquero, and Giovanni Villadoro for fruitful discussions. M.B. was supported by the DOE under Award Number DESC0007968. J.W.F. and B.R.S. were supported in part by the DOE Early Career Grant DESC0019225. This research used resources of the National Energy Research Scientific Computing Center (NERSC), a U.S. Department of Energy Office of Science User Facility located at Lawrence Berkeley National Laboratory, and the Lawrencium computational cluster provided by the IT Division at the Lawrence Berkeley National Laboratory, both operated under Contract No. DE-AC02-05CH11231. This research was supported by the Exascale Computing Project (17-SC-20-SC), a collaborative effort of the U.S. Department of Energy Office of Science and the National Nuclear Security Administration.

## Author contributions

M.B. coded and ran all simulations presented in this work, in addition to writing much of the analysis framework, and participated in all aspects of the project. J.F. was principally in charge of the physics analyses but also contributed to all aspects of the project. B.S. oversaw all aspects of the project and was principally in charge of writing. A.H. provided insight into analytic aspects of the physics. A.P., D.W., and W.Z. played key roles in helping adapt the AMReX code to this application.

## Competing interests

The authors declare no competing interests.
