## [Peer Review File · Nature Communications]

Reviewers' Comments:

Reviewer #1:

Remarks to the Author:

I read with great interest the paper entitled "Dark Matter with Axion Strings with Adaptive Mesh Refinement" by Bushmann et al.

This paper reports on recent numerical calculations of axion string dynamics in an expanding universe with a dedicated code based on the AMR technique. This paper is the culmination of a larger body of work done by the authors over the past years. This calculation is remarkable in the sense that it allows to predict the mass of this dark matter particle candidate quite accurately (assuming it is the only dark matter constituent). The methodology is sound, as it is based on a careful numerical experiment whose novelty lies in using adaptive mesh refinement to resolve the axion string core. This is particularly challenging since the string core is constant in size (in physical units) while the box is expanding.

I would recommend publication with some minor corrections.

1- The authors mention that the box size goes from 1200^3 Hubble volumes to 4 Hubble volumes between the initial and final times. If I understand correctly, it means that the box length goes from 1200 to 1.5 Hubble length. I think it is clearer and more honest to use the box length instead of the volume to state this.

2- Figure 2 caption: what is q_0 ? A typo?

3- Left column of page 3: the discussion on the fitting parameters is confusing. Sometimes the authors report the value of c_0 , sometimes only c_1 when comparing different papers. Why using c_{-1} and c_{-2} if they are never used.

4- On page 5 right column, the comment on domain wall stability in parenthesis is quite cumbersome. Either remove it or give more details.

5- First line of section 1 (AMR SIMULATION FRAMEWORK) does not make sense. "is" is probably missing.

6- In the discussion, I would like to see a comment on using patch-based AMR versus cell-based AMR. The latter seems to me more adapted to the problem at hand, as it can follow closely the complex shape of the strings.

7- METHODS: How exactly are equation 3 solved? Details are missing here. In general, the EOM are not characterized enough in the context of standard numerical methods. Are these equations hyperbolic? Are they solved using an operator split approach, with the wave part solved first and then the non-linear source terms second?

8- The authors mention the use of the CFL condition for stability. What is the wave speed for these EOM? It is not obvious directly from the equations that they have real eigenvalues when linearized. The laplacian term corresponds to waves moving at the speed of light, but what about the other terms? Are they becoming stiff?

9- The authors use SSPRK3 for the time integration. The need for SSP ODE solver makes sense in the context of TVD hyperbolic solver. It is not obvious why this applies for the EOM used here (see my remarks above).

10- Refinement criteria 2 and 3 are not justified. I guess they come from the particular form of the EOM but more details are needed.

Reviewer #2:

Remarks to the Author:

I was asked to give a general scientific opinion on the results presented in this manuscript, of their impact in the physics (particularly axion) community, and on the general scientific stand of this paper. I have never performed these kind of simulations and so I will avoid commenting on specific numerical aspects of the work.

The authors provide a new numerical calculation of the axion strings contribution to the axion abundance. This is an extremely relevant problem in the axion community. If axions are generated after inflation, it is in principle possible to calculate their cosmological abundance as a function of the mass and thus to extract the value of the mass associated with the DM abundance observed. A reliable value for this mass would be of extreme interest for experiments searching for DM axions (haloscopes). In particular, axion resonant cavities can achieve an excellent sensitivity to small axion couplings but require a very time-consuming scan of the mass region. Narrowing down the expected mass range would substantially accelerate this process.

The calculation of the axion abundance is, however, very difficult. One of the most outstanding challenges is the computation of the axion contribution from topological defects, in particular from axion strings. If this contribution is small, then axions would be produced mostly through the misalignment mechanism, which is numerically more accessible. However, if the string contribution is important, an estimate of the axion abundance becomes much harder. Different studies in the last four decades gave quite different (often, incompatible) results. The axion abundance from string decay depends on the axion radiation spectrum, which can be parameterized by a spectral index, q , which in general may depend on time. Many independent simulations by different groups provided different results for q and for its time dependence.

In my opinion, the work presented here represents a very important improvement toward the resolution of this outstanding problem. The authors perform a numerical simulation using an Adaptive Mesh Refinement (AMR) code to maintain high resolution around the string cores, where higher precision is required. Consequently, they achieve an effective resolution which would be unimaginable with static grids. Using this method, the authors conclude that the spectral index is roughly time independent and equal to 1. From this result, they argue that, in order for axions to constitute the totality of the DM their mass should be in the range $(40-180) \mu\text{eV}$.

The conclusion is not unreasonable. Arguments for a spectral index equal to 1 were presented also, for example, in Ref. [28]. The authors themselves give intuitive arguments to support their $q=1$ result in the Methods section of the present paper. I was somewhat surprised to see that the authors perform only one simulation. I would have expected a need for more simulations, perhaps with different initial conditions, to verify the stability of their result. In general, however, the authors did a remarkable and noteworthy job.

So, in my opinion this paper represents an important step toward understanding the dynamics of axion strings and the axion DM abundance. Furthermore, the paper is very well written and clear. My only concern about getting this result into Nature is that, in my opinion, the results are still controversial. In the recent literature we can find very different results, often incompatible among themselves and in disagreement with the results of this paper. The disagreement is not always small. As the authors mention, some studies find that much higher axion masses would be needed to saturate the DM component in the universe. The authors provide possible explanations for their disagreement with the other literature. I find their arguments reasonable. However, the disagreement is often based on choices such as how to fix χ_{UV} , or simply on resolution. Is this sufficient to rule out results that are significantly different? We may expect the resolution of these simulations to improve in the future. Should we expect, again, such substantial changes?

Besides these concerns, which should be discussed also in relation to the other referee's opinions, I want to underline again that this is a remarkable paper and an important step toward a better comprehension of the role of axion strings in the prediction of the axion DM abundance.

Reviewer #3:

Remarks to the Author:

This paper uses adaptive mesh refinement to simulate global $U(1)$ cosmic strings on a comoving lattice. A network of such strings may have been formed after inflation by a Peccei-Quinn scalar field, and could then persist to the QCD scale. The axion particle itself is the pseudo-Goldstone boson of the scalar field, and a prime dark matter candidate. Measuring the amount of pseudo-Goldstone radiation from this network of $U(1)$ strings therefore allows one to determine the resulting dark matter abundance. Understanding how the string network evolves and emits radiation is crucial. This will in turn allow accurate predictions of the resulting present-day axion mass (given other cosmological observables), permitting the viability of this scenario to be established.

The key innovation of the present work is the use of adaptive mesh refinement to simulate the network, allowing the string cores to be simulated with higher resolution than the overall network. This can allow larger overall simulation volumes to be studied, for longer simulation times. Using adaptive mesh refinement, the authors have demonstrated that the axion string network radiates energy with a scale-invariant spectrum.

Adaptive mesh refinement has been widely proposed for use in simulating cosmic string networks in the past, and so it is exciting to see it finally implemented and applied to a phenomenologically important research question. However, I believe that the Methods section should be expanded to include further numerical tests on the reliability of the simulation, particularly as this is the first paper by the authors to use this technique. A standard test for string network evolution simulations would be measuring the covariant conservation of energy within the simulation box. As the central claims of this paper relate to string radiation, I believe it is important to establish that energy (or a similar quantity) is globally conserved or that its loss is accounted for consistently.

Other numerical tests would also be desirable, for example a demonstration that the same results are obtained (perhaps for a smaller test simulation) with and without adaptive mesh refinement (i.e. zero refinement levels) given the same simulation code. Other valuable checks would include showing that the simulation results are independent of (amongst other things): increasing the number of refinement levels, further reducing the lattice spacing (what the authors call their 'comoving spatial difference'), and increasing the threshold string width for adding additional refinement levels. Again, these could be established for smaller simulation volumes over shorter periods of time in the Methods section.

In summary, then, this paper presents interesting new results for the evolution of a network of axion strings. The use of adaptive mesh refinement seems to allow a considerable improvement in dynamic range. My view is that the authors' central innovation is in their numerical methods, and the promise of increasing precision as the simulations are scaled up. Furthermore, the broader impact of this work lies in its potential applicability to other early universe simulation scenarios, although the paper as currently written is quite narrowly specific to axion strings and does not give the wider audience guidance as to the reliability of adaptive mesh refinement simulations. In any case, I would encourage the authors to expand their Methods section to include greater discussion of their numerical tests.

Response to Referees: “Dark Matter from Axion Strings with Adaptive Mesh Refinement”

Malte Buschmann, Joshua W. Foster, Anson Hook, Adam Peterson,
Don E. Willcox, Weiqun Zhang, and Benjamin R. Safdi

December 12, 2021

Dear Editor:

We want to thank the referees for taking the time to read through our paper and provide thoughtful and instructive feedback. They bring up a number of points, which we address below.

Response to Reviewer 1

Below, we address the specific comments and suggestions from Reviewer 1. Note that changes made to the manuscript in response are marked in red.

1- The authors mention that the box size goes from 1200^3 Hubble volumes to 4 Hubble volumes between the initial and final times. If I understand correctly, it means that the box length goes from 1200 to 1.5 Hubble length. I think it is clearer and more honest to use the box length instead of the volume to state this.

We changed to notation in the main text as suggested by the referee.

2- Figure 2 caption: what is q_0 ? A typo?

We thank the referee for pointing this out this typo. We corrected it to c_0 .

3- Left column of page 3: the discussion on the fitting parameters is confusing. Sometimes the authors report the value of c_0 , sometimes only c_1 when comparing different papers. Why using c_{-1} and c_{-2} if they are never used.

We agree with the referee that the notation is a bit confusing. Previous works by e.g. Gorghetto et al. [1] only quote the leading term c_1 , despite including c_0 in their fit, so we were unable to report their value for c_0 (or c_{-1} , c_{-2}). The reason for using c_{-1} and c_{-2} in the first place is that while these terms are irrelevant for the extrapolation to large \log 's, they are important at early times. Including them in the fit allows us to extend the fitting range to below $\log \sim 7.5$. However, the exact numerical values of these parameters are of little interest. As the functional shape of ξ is purely phenomenological testing both version (with and without c_{-1} , c_{-2}) over different fitting ranges serves as a good test for the stability of the leading term c_1 . We modified the text for clarity and removed references to numerical values of c_0 .

4- On page 5 right column, the comment on domain wall stability in parenthesis is quite cumbersome. Either remove it or give more details.

We agree with the referee. We moved the comment to the Method section and expanded on it.

5- First line of section 1 (AMR SIMULATION FRAMEWORK) does not make sense. "is" is probably missing.

We thank the referee for pointing this out, we fixed the missing "is".

6- In the discussion, I would like to see a comment on using patch-based AMR versus cell-based AMR. The latter seems to me more adapted to the problem at hand, as it can follow closely the complex shape of the strings.

Several aspects of the dynamics of our system make patch-based AMR a well-suited technique for the problem at hand. As the referee notes, the strings have complex shapes, causing the patch-based algorithm to refine a much larger volume than is occupied by the strings. However, a buffer region around the string must also be highly resolved as the string locations and configurations are themselves dynamical. This buffer region is naturally provided by the patch-based scheme, and we have chosen the buffer region to be sufficiently large in order to reduce the required frequency of regriding, which has considerable computational cost. Furthermore, we must not only resolve the strings but also the small-scale modes they emit, and the buffer region around the strings enables us to resolve, identify, and further track those small-scale modes with additional refinement criteria. While it is possible that there exists a suitable cell-based AMR scheme for simulating axion string dynamics, these inbuilt advantages of patch-based AMR coupled with the comprehensive features provided by AMReX led us to choose it for this work.

7- METHODS: How exactly are equation 3 solved? Details are missing here. In general, the EOM are not characterized enough in the context of standard numerical methods. Are these equations hyperbolic? Are they solved using an operator split approach, with the wave part solved first and then the non-linear source terms second?

The equations of motion are strongly hyperbolic and are thus stable to numerical evolution using explicit Runge-Kutta time integration techniques, provided a reasonable Courant factor is chosen. We have chosen to reduce the equations of motion (Eq. 3) to first order in time derivatives, substituting the conjugate momentum $\Pi_{1,2} \equiv \psi'_{1,2}$. We have added a few lines in the text further clarifying our numerical approach.

8- The authors mention the use of the CFL condition for stability. What is the wave speed for these EOM? It is not obvious directly from the equations that they have real eigenvalues when linearized. The laplacian term corresponds to waves moving at the speed of light, but what about the other terms? Are they becoming stiff?

The waves propagate at the speed of light, which corresponds to a unit velocity in simulation units. Linearizing the equations of motion about the solution

$(\psi_1^2 + \psi_2^2) = 1$ reveals these to be simple wave equations with real eigenvalues, and the other terms do not produce stiffness.

9- The authors use SSPRK3 for the time integration. The need for SSP ODE solver makes sense in the context of TVD hyperbolic solver. It is not obvious why this applies for the EOM used here (see my remarks above).

The choice of SSPRK3 was motivated by the joint considerations of solver accuracy and memory cost. Previous studies of axion string dynamics have implemented the second-order leapfrog algorithm, which we intended to improve upon using a higher-order integration scheme. AMReX currently implements the third-order SSPRK3 and the fourth-order RK4, and though RK4 would have provided potentially greater solver accuracy, its cost in memory would have been too great to run the simulation as described. We emphasize that the choice of SSPRK3 was purely motivated by choosing a higher-order solver available in AMReX rather than for its strong stability-preserving property.

10- Refinement criteria 2 and 3 are not justified. I guess they come from the particular form of the EOM but more details are needed.

We added some extra clarification to the text. In short, certain string dynamics, in particular strings that are reconnecting, can cause large shock waves that travel away from strings. The typical scale of these wave fronts is related to the string width and they therefore requires good spatial resolution as well. The two extra refinement criteria track these wave fronts in the two field components as they would not be otherwise captured by the first refinement criteria.

Response to Reviewer 2

Below, we address the specific comments and suggestions from Reviewer 2. Note that changes made to the manuscript in response are marked in red.

I was somewhat surprised to see that the authors perform only one simulation. I would have expected a need for more simulations, perhaps with different initial conditions, to verify the stability of their result. In general, however, the authors did a remarkable and noteworthy job.

First, we thank the referee for their kind words. We agree with the referee that multiple simulations with different initial states would be useful to diagnose the sensitivity to the initial state. We originally did not perform multiple simulations because they are computationally costly; we chose to put all of our resources towards one, large simulation. However, we have now performed a few such tests, as suggested by the referee, which are summarized in Supp. Figs. S11 - S14, with corresponding discussions in Methods Secs. F and G. The tests described in Sec. G are particularly relevant (Figs. S13 and S14). For these tests, we vary the number of modes entering the initial state, considering more and less modes (relative to the box size) compared to our fiducial simulation. We perform simulations on two statistical realization of each mode number choice in order to minimize the statistical uncertainty and further highlight possible systematic effects. However, given that this required four new simulations, we were forced to simulate using smaller box sizes and were only able to evolve out to $\log \sim 7$. Still, we hope that the referee finds these new tests, which themselves were still computationally expensive, to be useful. Interestingly, the results in Figs. S13 and S14 support the hypothesis that the string length per Hubble approaches a scaling solution (as claimed in [2]), robust to the initial state. The results also indicate that the spectral index, as measured at large \log , is not sensitive to the number of initial modes, at least to within the precision probed in this work. Of course, one limitation of these tests is that we were not able to evolve to larger \log values because of the computational resources involved. Still, we believe these new tests further support our approach, and so we thank the referee for the suggestion.

My only concern about getting this result into Nature is that, in my opinion, the results are still controversial. In the recent literature we can find very different results, often incompatible among themselves and in disagreement with the results of this paper. The disagreement is not always small. As the authors mention, some studies find that much higher axion masses would be needed to saturate the DM component in the universe. The authors provide possible explanations for their

disagreement with the other literature. I find their arguments reasonable. However, the disagreement is often based on choices such as how to fix x_{UV} , or simply on resolution. Is this sufficient to rule out results that are significantly different? We may expect the resolution of these simulations to improve in the future. Should we expect, again, such substantial changes?

It is true that recent results in literature have often produced very different results. Indeed, for decades now there has been a debate in the literature about the answer to this question: what is the axion mass that gives the correct DM abundance when the PQ symmetry is broken after inflation? It is partially because of this debate and the outstanding nature of this question that we feel this paper will be of interest to the broader Nature Communications readership. However, we want to stress that our results provides a substantial improvement over previous works, including the recent ones, with a dynamic range three orders of magnitude larger than other studies. Regarding cutoffs like x_{UV} , we provide various systematic tests showing the stability of our result. In terms of spatial resolution issues, a very recent paper [3] performed systematic tests which show that our string resolution appears to be sufficient, whereas those in [1, 2] suffer at late times.

Of course, it remains a possibility that the axion mass prediction is going to change further in the future, though we would be very surprised, given our results, if it moves substantially from our predicted window. This is because our increase in dynamic range allows us to do a suite of tests and robust analyses that simply were not possible with smaller simulation volumes. Even if the axion mass prediction does not move outside of our window, however, our paper will surely not be the final word on the subject. Future works will be needed to refine the mass prediction. However, what we can say for sure is that our work presents a game-changing approach to simulating the axion dynamics, which we show allows for a substantial and qualitative improvement in the dynamic range. This alone, in our opinions, warrants publication in Nature Communications.

Response to Reviewer 3

Below, we address the specific comments and suggestions from Reviewer 3. Note that changes made to the manuscript in response are marked in red.

I believe that the Methods section should be expanded to include further numerical tests on the reliability of the simulation, particularly as this is the first paper by the authors to use this technique. A standard test for string network evolution simulations would be measuring the covariant conservation of energy within the simulation box. As the central claims of this paper relate to string radiation, I believe it is important to establish that energy (or a similar quantity) is globally conserved or that its loss is accounted for consistently.

We appreciate the referee's concern about including further tests to establish the validity of the simulation. Indeed, we have included multiple additional tests, as described in our response to the previous referee and our response to the next comment. However, we believe that there are some issues that make it difficult to perform the precise test requested by the referee here, though we have performed a related test. Importantly, energy is not conserved because of the space-time cosmological expansion. Moreover, our cosmology includes contributions that redshift like matter and contributions that redshift like radiation. Thus, we cannot simply multiply the energy density by a simple function of redshift and get a conserved quantity. What we can do, on the other hand, is check that various components of the solution, whose properties we understand theoretically, are redshifting properly. An important test along the lines requested by the referee is currently provided in Fig. S2 described at further length in Sec. D of the methods. As the string network evolves, energy from the strings is transferred via radiation into axion and radial mode excitations, and accurately performing simulations should satisfy that the instantaneous rate of change in the total string energy density is equal and opposite to the instantaneous rate of change in the total energy density excluding the strings. We have evaluated the string tension (the linear energy density) of strings by determining the total energy density excluding the regions in the vicinity of strings per total length of strings in the simulation volume. Critically, the string energy density and corresponding emission rate into axion and radial mode excitations demonstrates the expected time-dependent scaling and agrees with the theoretically expected magnitude at the (statistics-limited) few percent level. We believe that this is a highly non-trivial check of both energy conservation (in the appropriate sector, where we understand it theoretically) and that the strings have the expected tension from theoretical arguments. We have added additional comments to the Methods section D to highlight how Fig. 2 may be interpreted as a test of the numerical procedure and possible energy leakage.

Other numerical tests would also be desirable, for example a demonstration that the same results are obtained (perhaps for a smaller test simulation) with and without adaptive mesh refinement (i.e. zero refinement levels) given the same simulation code. Other valuable checks would include showing that the simulation results are independent of (amongst other things): increasing the number of refinement levels, further reducing the lattice spacing (what the authors call their ‘comoving spatial difference’), and increasing the threshold string width for adding additional refinement levels. Again, these could be established for smaller simulation volumes over shorter periods of time in the Methods section.

We agree with the referee that these are important tests for convergence. We have added a section to Methods (Sec. F) describing an additional systematic test on a smaller simulation volume. In summary, we compare a simulation with a high-resolution static grid to a simulation with adaptive mesh refinement. Other than the grid layout both simulations are identical and based on the same initial state for maximum comparability. Therefore, the static grid serves as a base line since the entire field is over-resolved at all times. Note, however, that we are only able to evolve to $\log \sim 6$ because of the static grid.

We compute our quantities of interest in both cases, string length ξ and the instantaneous spectrum (at the final state). We find that the relative difference in string length between both simulations is less than 0.5% without any observable drift away from zero (Fig. S11), which is far subdominant compared to the statistical uncertainty we had assigned to ξ . Given that we are at a relatively low \log , we are not able to perform a fit to measure the spectral index q for the instantaneous spectrum. However, comparing the spectra themselves (Fig. S12) we see that the differences are again less than a percent over the range of k of interest, again indicating that differences between the AMR and static grid simulations are subdominant compared to statistical noise (as described more fully in Sec. F of Methods).

Together with the low-resolution test where we changed the number of refinement levels (this test was already described in our manuscript), we believe that this new test helps support the conclusion that as long as the strings are sufficiently resolved we can reproduce the results of interest independent of whether an AMR technique is used or a static grid. As this conclusion was to be expected, given that AMR is a well established technique, it is reassuring to see it play out in practice.

We thank the referees again for their time reviewing our manuscript and their constructive feedback. We believe that we have now addressed the points brought up by the referees, and we hope that our paper is now deemed ready to proceed towards publication.

Best,

The authors

References

- [1] M. Gorghetto, E. Hardy, and G. Villadoro, “More axions from strings,” *SciPost Phys.*, vol. 10, no. 2, p. 050, 2021.
- [2] —, “Axions from Strings: the Attractive Solution,” *JHEP*, vol. 07, p. 151, 2018.
- [3] J. R. C. C. Correia and C. J. A. P. Martins, “Quantifying the effect of cooled initial conditions on cosmic string network evolution,” *Phys. Rev. D*, vol. 102, no. 4, p. 043503, 2020.

Reviewers' Comments:

Reviewer #1:

Remarks to the Author:

I am happy with the modifications made by the authors.

I think the paper is now ready for publication.

Reviewer #2:

Remarks to the Author:

I am satisfied with the author's responses to my comments and appreciate the additional tests performed by the authors. I understand the difficulty and time required for such large numerical simulations.

In my opinion, the justifications given about the choice of the parameters are satisfactory.

I maintain my opinion that this is an important improvement with respect to the previous analyses and I am favorable to the publication of these results in Nature Communication.

Reviewer #3:

Remarks to the Author:

The authors have added substantial discussion to the Methods section providing valuable tests of the reliability of their simulations.

They have not carried out the test of energy conservation that I suggested. To be specific, it would have been good to compare the left and right hand sides of the Friedmann equation during the simulation evaluated numerically from lattice quantities such as the energy density and pressure. This should indeed have been possible with their cosmology. Nevertheless the additional tests and comparisons they have carried out are satisfactory reassurance to me of the reliability of their results.